# Formulation and In-Vitro/Ex-Vivo Characterization of Pregelled Hybrid Alginate–Chitosan Microparticles for Ocular Delivery of Ketorolac Tromethamine

**DOI:** 10.3390/polym15132773

**Published:** 2023-06-21

**Authors:** Zeinab Fathalla, Adel Al Fatease, Hamdy Abdelkader

**Affiliations:** 1Department of Pharmaceutics, Faculty of Pharmacy, Minia University, Minia 61519, Egypt; zianab.mohamed@minia.edu.eg; 2Department of Pharmaceutics, College of Pharmacy, King Khalid University, Abha 62223, Saudi Arabia; afatease@kku.edu.sa

**Keywords:** ketorolac, hybrid microparticles, permeation, chitosan, alginate, lens cell lines

## Abstract

Innovative hybrid chitosan–sodium alginate (Ch–Ag) microparticles (MPs) were fabricated using both the ionic gelation method as well as the pre-gelation technique. The hybrid Ch–Ag MPs were studied for size, zeta potential, morphology, mucoadhesion, in-vitro release, corneal permeation, and ocular irritation using lens and corneal epithelial cell lines. The average particle size ranged from 1322 nm to 396 nm. The zeta potential for the prepared formulations showed an increase with increasing Ch concentrations up to a value of >35 mV; the polydispersity index (PDI) of some optimized MPs was around 0.1. Compared to drug-free MPs, ketorolac-loaded Ch–Ag MPs demonstrated a drug proportion-dependent increase in their size. SEM, as well as TEM of KT-loaded MPs, confirmed that the formed particles were quasi-spherical to elliptical in shape. The KT release from the MPs demonstrated a prolonged release profile in comparison to the control KT solution. Further, mucoadhesion studies with porcine mucin revealed that the KT-loaded MPs had effective mucoadhesive properties, and polymeric particles were stable in the presence of mucin. Corneal permeation was studied on bovine eyes, and the results revealed that Ch-based MPs were capable of showing more sustained KT release across the cornea compared with that for the control drug solution. Conclusively, the cytotoxicity assay confirmed that the investigated MPs were non-irritant and could confer protection from direct drug irritation of KT on the ocular surface. The MTT cytotoxicity assay confirmed that KT-loaded MPs showed acceptable and reasonable tolerability with both human lens and corneal epithelial cell lines compared to the control samples.

## 1. Introduction

The distinctive anatomical nature of the eye and the drug elimination mechanisms of nasolacrimal drainage, reflex tearing, and eye blinking make topical delivery through the eye a challenging route of administration [1,2]. The ocular bioavailability after topical administration of eye drops for hydrophilic drugs is typically less than 1%. Over many years, eye drops have shown prominent advantages for many reasons. They are non-invasive, easy to use, and have simple formulation methods and cost-effective dosage forms. The advent of non-steroidal anti-inflammatory drugs (NSAIDs) in ophthalmology and repeated administration of NSAID eye drops have caused several ocular side effects and eye discomfort to be reported with the conventional mode of administration [3,4]. Further, frequent dosage administration impairs patient compliance, which results in modest treatment success [5]. Ketorolac tromethamine (KT) is a commonly used NSAID eye drop for treating allergic conjunctivitis and post-operative pain and inflammation after cataract surgeries. Only a small fraction of the instilled dose (5 mg/mL) can permeate through the ocular barriers and cross into the anterior chamber and the human lens. Ophthalmologists prescribe KT eye drops on a repeated dose regimen; however, this mode of treatment has been associated with ocular side effects, including burning, stinging, and conjunctival irritation [6].

Polymeric particles have attracted great attention because they have many benefits compared with conventional ocular eye drops, such as sustained release, prolonged residence time on the ocular surface, and enhanced ocular barrier penetrating characteristics [7]. In addition, polymeric nano-systems exhibit enhanced mucoadhesion to the ocular surface and resist rapid washout and dilution from tears [7].

Different types of polymeric nano-carrier systems have been extensively investigated for the ocular delivery of many drugs. Alginate-based microparticles are considered a consequential, important, and valuable system. The exceptional characteristics of alginate nano-systems include their abundant availability, non-toxicity, biodegradability, and sustained release patterns [8,9]. Moreover, sodium alginate (Ag), an anionic hydrophilic polymer, has excellent characteristics for controlling the release of a variety of drugs. However, the prospective use of alginate-based nano-carriers revealed a poor sustainable drug release pattern due to high erosion rates from these Ag-based hydrogel matrices. In order to improve the sustained release efficiency of Ag-based systems, an oppositely charged polymer, such as the cationic polymer chitosan, has been investigated with Ag-based systems [9].

Alginate MPs can be formulated simply through induction of pre-gelation and cross-linking using calcium ions. This study utilized the straightforward gelling characteristics of sodium alginate through alginate pre-gelation. Then, an aqueous polycationic polymer solution of chitosan was added to form polyelectrolyte complex hybrid MPs. This hypothesis was previously investigated with poly-L-lysine and alginate for sustained delivery of biological macromolecules [10]. Immunogenicity and toxicity were reported with the use of the synthetic polylysine cationic polymer (PLL) [11]. The natural cationic polymer chitosan (Ch) combined with the anionic alginate polymer has attracted growing interest. This is due to chitosan’s safety, biocompatibility, and capacity to form complementary anionic–cationic (polyelectrolyte) complexes for sustained drug delivery [12].

These nano-materials composed of polymers of natural origin have also been used as vehicles for controlling and prolonging the delivery of a wide range of therapeutics. Chitosan can be considered a natural and safer alternative than PLL [12]. In addition, the most convenient natural polymer for forming stable polyelectrolyte complexes with chitosan (Ch) is the anionic polymer Ag [13,14]. This study aimed to prepare hybrid alginate–chitosan polyelectrolyte microparticles loaded with the NSAID ketorolac in an attempt to prolong its release through the corneal barriers; further, mucoadhesion and ocular safety were investigated on corneal and lens epithelial cell lines. 

## 2. Materials and Methods

Ketorolac tromethamine (KT), chitosan (Ch) (viscosity 20,000 mPa.s), sodium alginate (Ag), calcium chloride, and mucin from a porcine origin were bought from Sigma Co., London, UK.

### 2.1. Generation of Ch–Ag MPs Using the Modified Ionic Gelation Method

Ch–Ag MPs were formed by employing the modified ionic gelation method [15,16]. Different amounts of chitosan (Ch) were dissolved in diluted acetic acid (0.5%) at seven different concentrations: 0.03, 0.04, 0.05, 0.06, 0.08, 0.1, and 0.2% and the pH was adjusted to pH 5.5. Sodium alginate (Ag) was dissolved in ultrapure water to obtain two concentrations, 0.05% and 0.075%. Then, all the solutions were filtered through 0.45 µm filters. The MPs were harvested by the slow addition of Ag to the aqueous solutions of Ch under magnetic stirring (700 rpm) for 30 minutes (min).

Some selected Ch-Ag MPs were prepared using different Ch-Ag volume ratios: 1:1, 2:1, 2.5:1, and 3:1 *v*/*v*. In addition, various concentrations of Tween 80 (a stabilizer) were also investigated. 

### 2.2. Preparation of Ch-Ag MPs Containing Calcium Chloride Using Ionic Pre-Gelation Method

The hybrid Ag-Ch polyion complex MPs were prepared using the ionotropic pregelation of the anionic Ag polymer with the cross-linker calcium chloride (CaCl_2_) and subsequent addition of the cationic Ch according to the previously published methodology [17].The method was based on polymeric matrix formation without the involvement of heat. The pregelation of the anionic polymer Ag was formed in the presence of Ca^2+^ ions; then, the cationic polymer Ch was added to induce polyelectrolyte complexation [18]. The anionic polymer Ag was pregelled at pH 5.3; Ch was separately dissolved in acetic acid (0.5%) and the pH was raised to 5.5 [17]. Then, CaCl_2_ solution with a concentration of 0.22% *w*/*v* was added dropwise to 5 mL Ag solution (0.063% *w*/*v*) while stirring for 10 min at 700 rpm at ambient conditions. Then, 5 mL Ch solution (0.095% *w*/*v*) was mixed with the Ca-Ag pre-gel under magnetic stirring for 30 min. The final Ch:Ag ratio was 1.5:1 *w*/*w*. The formed MPs were purified using ultracentrifugation (16,000× *g*/30 min) at 4 °C. Finally, the MPs were redispersed in water to measure the size and zeta potential. Different Ch to Ag mass ratio have been used (1:1, 1.5:1, 3:1, 5:1, 1:1.5, 1:3, and 1:5) *w*/*w*, respectively.

#### 2.2.1. Effect of Processing Variables on Particle Sizes

##### Effect of Calcium Chloride Concentration and Stirring Time

Different concentrations of CaCl_2_ were studied, i.e. 0.1%, 0.22%, 0.43%, and 0.87% (*w*/*v*). Effects of changing stirring time (0.5, 1, 2, 5 and 8 h) were also investigated.

##### Effect of pH of Sodium Alginate

The effect of changing the pH of Ag solutions on the morphological characteristics of prepared microparticles was investigated. Different pH values were used (2.6, 5.3 and 7.3). After optimizing all the conditions required to obtain satisfactory formulations, selected ratios with optimum conditions were chosen for loading with KT in order to formulate KT-loaded MPs for release study. The MPs loaded with the drug, their EE% was calculated. KT was dissolved in Ch solution (pH 5.5) and the cryoprotectant mannitol with a concentration of 5% was added before starting the formulation process in order to prevent the aggregation of MPs. 

### 2.3. Estimation of Entrapment Efficiency (EE%)

The EE% of Ch-Ag based MPs was assessed as follows: the prepared MPs were purified using the ultracentrifugation at 12,000× *g* for 30 min at 4 °C. The drug content was analyzed spectrophotometrically at 314 nm [19] and EE% was determined through Equation (1):(1)$$Entrapment\;Efficiency\left(EE\%\right)=\frac{\left(T_{p}-T_{f}\right)}{T_{p}}\times 100$$
where T_p_ is the amount of drug used initially to prepare the microparticles and T_f_ is the amount of drug in the supernatant [20,21].

### 2.4. Particle Size, PDI and Zeta Potential Measurements

Malvern Zetasizer 3000HSA (London, UK) was used to measure the sizes, polydispersity index (PDI) and zeta potential at 25 °C after appropriate dilution with distilled water. 

### 2.5. Morphological Investigation Using SEM and TEM

For SEM, KT-loaded MPs were imaged using a Zeiss EVO 50 (Cambridge, UK). An air-dried, sputter-coated drop of the formulation dispersion was added on 1 cm glass slip that was attached to an aluminum stub. The transmission electron microscope (TEM) of the type (JEOL, JEM 1010, Tokyo, Japan) with a voltage of 80 kV was used to image the selected MPs. One drop of the sample after appropriate dilution was instilled onto a copper grid that was coated with carbon, and left to form a thin film before being stained with a drop of a 2% solution of ammonium acetate in ammonium molybdate (2% *w*/*v*) solution. The stained samples were imaged using JEOL software, Tokyo, Japan. At 20.00 kV. 

### 2.6. DSC

Using Mettler Toledo DSC 822e0, Greifensee, Switzerland, the thermal analyses of Ch, Ag, drug, physical mixtures (PM) and the lyophilized powder of the drug-loaded MPs were performed. In separate aluminum pans, the samples were weighed while the temperature of the pans was steadily raised from 25 to 300 °C at a rate of 10 °C/min. The flow rate used to purge nitrogen gas was 45 mL/min.

### 2.7. FTIR 

The FTIR spectra of the aforementioned samples were studied using an FTIR spectrometer (Thermo Scientific iS5, Bartlesville, OK, USA). The samples were applied in precise concentrations (2–4 mg) to create a thin layer that covered the diamond window. The FT-IR spectra were captured with 120 scans at a rate of 2 cm^−1^. Using Omnic software version 8.2, New York, NY, USA); the data was gathered and analyzed.

### 2.8. In-Vitro Drug Release Study

The Franz diffusion cells were used to study in-vitro release. Phosphate buffer saline (PBS) pH 7.4 was the release medium and transferred into the receptor chambers, which were then constantly agitated with tiny magnetic bars. A semipermeable membrane (12–14 kDa) separated the donor and receptor chambers. The temperature was set at 35 °C. Each formulation was placed into the donor domain with two milliliters. At regular intervals, samples (1 mL each) were taken out and compensated with a similar volume of the fresh media. The amount of drug released was determined spectrophotometrically at 314 nm.

### 2.9. Mucoadhesion Measurements

The stability and mucoadhesion of Ch-Ag MPs in the presence of mucin were investigated through incubation of the MPs with mucin. Two different in-vitro techniques were employed to evaluate the stability and interactions of KT loaded MPs with mucin. The first technique was about measuring the viscosity of mucin (0.04% *w*/*v*) solution before and after incubation at 35 °C in the presence of Ch based MPs or Ch solution alone. The viscosity was measured using a Brookfield viscometer.

The second approach was used to investigate how mucin affected both the loaded microparticles’ zeta potential and the zeta potential of Ch solution alone. The samples were incubated at 35 °C with a moderate agitation (200 rpm) in the mucin solution. The zeta potential of the samples was assessed at predetermined intervals (30, 60 and 120 min) throughout the incubation process. Additionally, before incubation, equal amounts of mucin solution (0.4 mg/mL) and Ch solution were vortexed for 1 min together with the MPs. The zeta potential of the mixtures was then determined using the Zetasizer [22].

### 2.10. Ex-Vivo Permeation (Transcorneal Permeation) Studies

Bovine eyeballs freshly collected from a nearby abattoir served as the source of the corneas used in the ex-vivo assay as previously reported [1]. The Franz diffusion cells were employed and the dissected tissue was positioned with the endothelium side towards the receptor compartment between the donor and receptor compartments. The receiving compartment was filled with PBS (12 mL, pH 7.4); while a sample of 2 mL of drug solution or KT MPs equivalent to 5 mg/mL KT (drug dose) was added to the donor domain. The pure drug solution was used as a control. The temperature was kept at 35 °C ± 1 °C [23]. Samples (1 mL) were taken out of the receiving solution and compensated with new PBS to make up for the withdrawn sample volume. The extent to which the drug crossed the cornea was determined as above-mentioned.

The permeation parameters: P_app_, and lag time t_L_ were estimated through Equation (2) [24].
(2)$$P_{app}=Slope\;\frac{1}{ACo}$$
where the slope or the flux was estimated from the slope of amount of KT permeated versus time curve; t_L_ was estimated from the intercept; C_o_ was KT concentration (µg/mL) in the donor domain initially added and A was surface area (cm^2^).

### 2.11. MTT Reduction Cytotoxicity Test on Human Primary Corneal Epithelial Cells as Well as Human Lens Cells

The MTT cytotoxicity test was performed on two types of cells: B-3 human lens epithelial cells (ATCC CRL-11421) and human corneal epithelial cells (ATCC pcs-700-010). Both types of cells were seeded (2 × 10^4^ cells/well) into 96 well plates using Eagle Minimum Essential Medium growth media containing 20% Fetal bovine serum (LGC standards) for B-3 human lens cells; hydrogen peroxide (H_2_O_2_) was used as a reference strong irritant control. Corneal Epithelial Cell Basal Medium containing apotransferrin, hydrocortisone hemisuccinate, L-glutamine, rh insulin and CE growth factor was used as growth media for corneal cells [25]. 

The two tested formulations chosen were both drug free and drug loaded (Ch1:Ag0.5) and (Ch1.5:Ag1) MPs. The growth media served as the negative control, while benzalkonium chloride (BKC) 0.01% *w*/*v* was employed as the strong irritant control [25]. After 4 h and 24 h of treatment, the medium was drained, and the cells were twice rinsed with sterile PBS at 37 °C. The cells were subsequently incubated for 4 h at 37 °C in Corneal Epithelial Cell Basal Medium (LGC standards) with 200 µL of a 0.5 mg/mL MTT solution per well. The MTT solution was removed after incubation, and sterile PBS was used to wash the wells twice. Two hundred µl of dimethyl sulfoxide was dropped into each well to dissolve the dye and analyze it at 540 nm.

### 2.12. Statistical Analysis

A one-way ANOVA was used for the statistical analysis, and a *p* value between 0.05 and 0.001 was regarded as statistically significant. San Diego, CA, USA; Graph-Pad Software Version 3.05 was used for these analyses. A pair-wise Tukey’s comparison with a 95% confidence level was carried out.

## 3. Results and Discussion

In order to prepare Ch-Ag micro-reservoir systems using the ionotropic gelation process, two aqueous polymeric solutions had to be combined at ambient temperature. Preliminary studies were conducted in order to identify the proper concentrations of Ch and Ag that enabled the production of the colloidal dispersions of MPs but not aggregates due to the increased viscosity of Ch solution. The final concentration ranges for Ch and Ag were 0.03–0.2% *w*/*v* and 0.05–0.075% *w*/*v*, respectively, for the optimization study. Similar results have been reported elsewhere [26]. 

### 3.1. Effects of Processing Parameters on Particle Size Measurements

#### 3.1.1. Effect of Changing Polymer/Polymer Ratios on Size and Zeta Potential

Table 1 and Table 2 show the effects of changing concentrations of either Ch and/or Ag on the average size of drug-free NPs. It could be seen that increasing Ch concentrations led to decreases in particle size, but this was up to a certain limit. After that, the size started to increase, as seen in Table 1. Upon increasing Ch concentrations from 0.03% *w/v* to 0.08% *w/v*, the size decreased significantly (*p* < 0.001) from 1322.4 ± 18.4 nm to 396.2 ± 6.40 nm. However, upon increasing Ch concentrations up to 0.2% *w/v*, there was a remarkable (*p* < 0.05) increase in the size (524 ± 11.10 nm). Increasing Ag concentrations from 0.05% *w/v* to 0.075% *w/v* at constant Ch concentration (0.05% *w/v*) led also to significant increases (*p* < 0.001) in the average size from (402.3 ± 4.52 nm) to (508.4 ± 27.8 nm), respectively, Table 1 and Table 2. This might be related to the higher concentration of both polymers and the availability of COO- and NH4+-charged groups from Ag and Ch, respectively, and thus, more MPs were formed during the reaction. In addition, the decrease in size of MPs until a certain concentration followed by an increase could be attributed to the interaction between the oppositely charged polyionic polymers in stoichiometric proportion, after which there was minimal reaction between two polymers. It was clear from the data in Table 1 and Table 2 that these ratios were 0.05%:0.05 and 0.05%:0.075% Ch: Ag, where the sizes were 402.3 4.52 nm and 508.36 27.8 nm, respectively. This result was in good agreement with that obtained by [26].

Zeta potential for prepared MPs seemed to increase with increasing Ch concentrations. This could be due to the greater availability of NH_3_ groups with raising the Ch concentration and increasing surface positive charge density on the formed MPs. For example, the zeta potential of (Ch0.3:Ag0.5) formulation, was 19.3 ± 0.21 mV while upon increasing Ch concentration, the zeta potential increased significantly (*p* < 0.01) to be 38.7 ± 1.01 mV with Ch2:Ag0.5 formulation. On the other hand, PDI for all prepared formulations whose size was less than 1µm was less than 0.5, which seemed to be within the acceptable range. For example, PDI for ratio 0.05:0.075 *w*/*v* (Ch:Ag) was 0.266 ± 0.003, Table 2. This confirmed the uniformity of the prepared MPs.

Changing the volume ratio during preparation of Ch: Ag MPs has affected the size and this may be due to the stoichiometric ratio between Ch and Ag. With formulation (Ch0.5:Ag0.5), when the volume ratio was 1:1 *v*/*v* (Ch:Ag) the size was in 1344 ± 31.95 (PDI: 0.974) while upon increasing the volume ratio up to 3:1 *v*/*v* (Ch:Ag), the size decreased significantly (*p* < 0.001) to 384.6 ± 18.81 nm (PDI: 0.052). The same was also true for the ratio (0.2:0.05) Ch:Ag, Table 3. 

#### 3.1.2. Effect of Tween 80 on Size Measurements and Zeta Potential

Tween 80 showed marked effects on the sizes of the generated MPs. Tween 80 is a non-ionic surfactant that could inhibit aggregation of the MPs; this could be attributed to improving wettability and decreasing interfacial tensions between the formed MPs. Data presented in Table 4 show that upon addition of Tween 80 with concentration (0.5% *w*/*v*) during the preparation process, the size of the MPs decreased significantly (*p* < 0.05) from 394.6 ± 2.628 nm to 343.9 ± 5.13 nm and from 528.36 ± 22.72 nm to 461.83 ± 6.76 nm for Ch0.5:Ag 0.5 and Ch2:Ag0.5 formulations, respectively. However, it seems that increasing Tween 80 concentration beyond this had insignificant (*p* > 0.05) effects on the size. Tween 80 (1% *w*/*v*) containing MPs gave an average size of 340.11 ± 6.88 nm for the same Ch0.5:Ag0.5 formulation. These findings suggested that Tween 80 had stabilizing effects and it prevented Ch aggregation during the preparation process as [27]. It is worth mentioning that the addition of Tween 80 to the three tested samples (Table 4) did not change the zeta potential significantly (*p* > 0.05) for the formed MPs as they retained their original values listed in Table 1.

#### 3.1.3. Effect of Different Concentrations of KT on Size Measurements

The results also indicated that the incorporation of KT into Ch-Ag MPs led to drug concentration-dependent increases in their sizes compared with plain (drug free) MPs (Table 5). For example, the size of (Ch0.5:Ag0.5) formulation for plain and loaded MPs with 5 mg/mL KT was 394.6 ± 2.628 nm and 497.90 ± 10.55 nm, respectively which was a significant (*p* < 0.05) increase in the size of loaded MPs. Further, doubling drug concentration from 5 to 10 mg/mL increased MPs’ sizes to 523.3 ± 5.76 nm for the same formulation ratio; however, the increase was not significant (*p* > 0.05), Table 6. This was in agreement with data that have been reported elsewhere [28]. However, the zeta potential of the Ch0.5:Ag0.5:KT5 formulation, Table 5, did not change significantly (*p* > 0.05) compared with the Ch0.5 Ag0.5:KT10 formulation, Table 6. The values were 26.7 ± 2.34 mV and 24.7 ± 0.88 mV, respectively. This could be attributed to that the fact that, less space was available for drug encapsulation due to the polymers that make up the majority of the MPs matrix. In addition, a high concentration of KT may increase the possibility of electrostatic interaction between KT and the polymers used [26].

#### 3.1.4. Effect of Different pH Values of Sodium Alginate Solution on Size Measurements

The effects of pH of Ag solution were studied and was found that pH 5.3 produced MPs of lower sizes. Most crucial for optimal interaction and the creation of the polyionic complex is the protonation of the amine group of Ch and the ionization of the carboxyl group of Ag and this what happened within this range. Similar results were reported were reported elsewhere; the generation of polyionic complexes-based MPs with lower particle sizes was favourably formed at pH of 4.7 [29]. Further, Ch is insoluble at neutral or alkaline pH and dissolution of Ch required acidic pH. Ch was less likely to be accessible for the creation of MPs when Ag solution with a higher pH was added because Ch is likely to precipitate out of solution [30]. When Ag solution with a neutral pH was added, the majority of Ch’s amino groups became unprotonated and were unable to interact ionically with alginate solution. By enabling a stronger interaction between Ch and Ag and causing the development of more compact with smaller sizes, using an Ag solution with a pH slightly lower (5.3) than normal overcomes these issues [26]. Figure 1 shows that the use of acidified Ag solution (pH 2.6) to prepare MPs results in a significant (*p* < 0.001) increase in the size of the formed MPs compared with the size obtained when the pH of Ag was 5.3 within this range [31].

Increasing the pH of Ag from 5.3 to 7.3 led to a remarkable (*p* < 0.05) increase in the size from 348.8 ± 2.00 nm to 421.5 ± 9.12 nm, (Figure 1). This could be due to the addition of Ag solution with a neutral pH resulting in the unprotonation of the amine groups of Ch and hence, this could undermine the ionic interactions with Ag. This could explain that formation of MPs with larger particle sizes [32].

It is also clear the zeta potential of the tested sample decreased significantly (*p*< 0.05) from 40.5 ± 1.4 mV to 26.13 ± 0.47 mV, (Figure 1) upon shifting the preparation pH from 2.6 to 5.3, respectively. This may be due to the fact that at acidic conditions, the Ch polymer is highly protonated, and upon increasing the pH of the Ag solution to 5.3, the carboxyl group of Ag is ionized and ready for reaction with the amino groups of Ch to form the MPs. However, a further increase in the pH of Ag has insignificant (*p* > 0.001) effect of the zeta potential of the MPs.

The preparation of Ch-Ag MPs using the pre-gelation method required to optimize the preparation conditions and a selected formulation will be further studied for drug release, physical stability, and corneal permeation study. So, with mass ratio (1.5:1) Ch:Ag and CaCl_2_ concentration of 0.22% *w*/*v*, stirring for 0.5 h would be the most appropriate conditions to prepare these MPs.

### 3.2. Entrapment Efficiency Measurements

Table 5 shows the EE% of Ch-Ag-loaded MPs. It was obvious that the EE% of all tested samples was between 19% and 24% when MPs were loaded with 5 mg KT. It was noticed that encapsulation of the drug was higher at the stoichiometric concentration of both polymers, Ch:Ag (0.05:0.5 mg/mL), while upon increasing polymer concentration, the EE% decreased. This may be due to the fact that at a constant concentration of Ag, KT may electrostatically interact with the Ch polymer whose concentration is increased. On the other hand, increasing the concentration of KT up to 10 mg leads to a significant (*p* < 0.01) increase in the EE% of the drug; for example, for the Ch0.5:Ag0.5 formulation, the values of the EE% were 24.53 ± 2.48% and 69.5 ± 2.13% with 5 mg and 10 mg KT, respectively, and the same is also true with loaded MPs in the Ch1:Ag0.5 formulation. Only two different ratios have been used to test the effect of KT concentration on the EE%, as these are the only ratios that gave an optimum particle size and PDI within the acceptable range, as their PDI is less than 0.9 and the particle size is within the submicoron-size range. Table 6 represents the effect of 15 mg/mL KT on the EE% of loaded MPs. It is obvious from the data that the loading efficiency of MPs tends to fall with further increases in the amount of drug, and this may be due to oversaturation and precipitation of KT during MPs preparation.

### 3.3. Effect of Different Formulation Parameters on Size, Zeta Potential and PDI of Pregelled MPs

#### 3.3.1. Effect of Different Polymer/Polymer Ratios

Table 7 shows the average particles size, zeta potential and PDI for Ch-Ag MPs containing 0.22% *w*/*v* CaCl_2_. Average particles size of the prepared MPs seemed to decrease significantly (*p* < 0.05) with increasing Ch:Ag ratio from (1:1) to (1.5:1) as the size was 449.33 ± 12.5 nm and 348.8 ± 2.0 nm, respectively. Also, the MPs prepared showed a narrow size distribution; for instance, the PDI for Ch1:Ag1 formulation was 0.109 ± 0.036 which confirmed the uniformity of formed MPs. The stoichiometric complexation between Ch and Ag was observed at ratio 1.5:1 because before or after this ratio, the size has been increased. However, there was a marked (*p* < 0.001) increase in the size of the MPs after further increases in the concentrations of either Ch or Ag as MPs recorded micro-range and also non uniformity of particles size distribution was noticed as PDI increased, Table 7. Meanwhile, the zeta potential of the formed MPs grew greater with increasing Ch concentrations. This could be ascribed to increasing the positive charge density available on the surface of the formed MPs that were contributed by the Ch molecules with using higher concentrations of the cationic polymer Ch.

#### 3.3.2. Effect of Different Concentration of Calcium Chloride

Increasing the cross-linking agent (CaCl_2_) concentrations led to a pronounced (*p* < 0.05) decrease in the sizes of the formed MPs, Table 8. For example, Ch1.5: Ag1 formulation recorded a size of 487.4 ± 1.90 nm at a low concentration (0.1% *w*/*v*) of CaCl_2_ and decreased significantly (*p* < 0.001) upon increasing CaCl_2_ concentration (0.87% *w*/*v*) to be 172.9 ± 4.08 nm, Figure 2. 

Knowing how alginate interacts with the divalent cations can help explain why particle size decreased when CaCl_2_ concentrations increased and reacted with Ag, from 0.1% to 0.87% *w*/*v*. Due to the “Zig-Zag” shape of the alginate molecules, the Ca^2+^ ions could bind to guluronic acid and mannuronic acid blocks with ease. Further, the pregel state was essential for enabling the ionic interactions between Ag and Ch to form MPs [32]. The cross-linker CaCl_2_ has to be present in small amounts (less than 0.2% mass ratio) in order to form the negatively charged MPs. Initially, the calcium alginate pregelled was encased in a positively charged Ch polymer. The cationic polymers Ch could inhibit further contribution to the binding between calcium and Ag ions [32]. These results are in contrast with those obtained by Chopra et al., who found that increasing cross-linker concentration led to increasing the average particle size of formed MPs [33].

#### 3.3.3. Effect of Changing Stirring Time

Changing the stirring time of the preparation process had an effect on the average size of the prepared MPs, (Table 9). Increasing stirring time led to a remarkable (*p* < 0.05) increases in the particle size, i.e., for formulation Ch1.5:Ag1, increases in the time from 0.5 h to 8 h results in an increment in the average size from 348.7 ± 2.00 nm to 480.6 ± 20.7 nm, respectively. This might be due to the fact that increasing stirring time led to the aggregation of the MPs and hence, increase in size. It could be concluded that 0.5 h was the optimum time required to prepare MPs using the pre-gel method as it also gave a narrow size distribution of MPs, PDI (0.121 ± 0.005) which indicated the uniformity of the formulations. Sarmento et al., confirmed these results as they reported that the average particle size was affected by time as he prepared Ch-Ag MPs loaded with insulin using the pregelation method; as the alginate-calcium intermolecular cross-links grew stronger over time, more contacts between alginate chains formed the pre-gel nucleus [34]. However, stirring times of 30 min and 24 h showed no significant effects on mean Ag-Ch MPs’ sizes; this could suggest instant formation of the MPs [31].

### 3.4. Effect of Different Concentrations of KT on Entrapment Efficiency

Table 10 shows the effects of different concentrations of KT on the EE% of the formed MPs. As seen from the data, the ratio 1:1 gave a higher EE% compared with the (Ch1.5:Ag1:KT10) formulation. This could be attributed to the fact that upon increasing the Ch/Ag ratio, aggregation of some MPs occurred, and thus, this was likely to decrease the capacity of those MPs to entrap drugs. It has been reported that loading effectiveness was negatively impacted by Ch concentrations because, at higher concentrations, Ch caused the formation of aggregates when added to Ag solution [26].

Increasing KT concentrations from 5 mg to 10 mg led to nearly two-fold increases in the EE% of the loaded formulation. For example, the EE% of the Ch1:Ag1 formulation at 5 mg KT was 51.44 ± 5.33% while upon increasing concentration to 10 mg, the EE% increased significantly (*p* < 0.001) to be 89.09 ± 1.66%. This may be due to the fact of more drugs are available in the media for the MPs to entrap because KT is a hydrophilic drug, rapidly dissolved and distributed in the solution. In addition, it was obvious that the drug concentrations affected both size and zeta potential measurements. The higher the drug concentration, the larger a size was recorded and these increases were significant (*p* < 0.05). On the contrary, the zeta potential decreased remarkably (*p* < 0.05) with increasing concentrations of KT. This could be due to the partial neutralization of negatively charged acidic drugs with the basic amino group residue of Ch polymer, and similar results were obtained with Gupta et al. [35]. 

### 3.5. Morphology and Structure Characterisation of Chitosan-Sodium Alginate MPs 

#### 3.5.1. SEM

The morphology of selected MPs was elucidated utilizing the SEM. The MPs appeared quasi-spherical-to-oval in shape (Figure 3). The surface of Ch-Ag loaded MPs appeared quite fluffy with an irregular surface, and the sizes were close to 0.5 µm.

There was no significant difference in the shape of the MPs of Ch-Ag and the one prepared using the pre-gelation upon carrying out the SEM analysis. Figure 4 indicates that KT loaded Ch-Ag (1.5:1 *w*/*w*) MPs containing CaCl_2_ (0.22%) appeared spherical in shape with an average particle size of 620.1 ± 7.22 nm and PDI (0.422 ± 0.05). 

#### 3.5.2. Transmission Electron Microscope (TEM)

To gain further insights into the morphological characteristics of the formed MPs, TEM was employed. Ch-Ag based MPs demonstrated spherical MPs with slightly rough surfaces (Figure 5).

Ch-Ag MPs with a 1.5:1 *w*/*w* ratio and prepared using the pregelation method, showed MPs with perfectly spherical in shape and good size distribution. The background structures of this image showed what looked like crystals, which may be related to the CaCl_2_ used in this formulation.

### 3.6. DSC 

The DSC thermograms of drug, Ag, PM and drug loaded (Ch1:Ag0.5:KT10) MPs are shown in Figure 6. The thermogram recorded for Ag exhibited a broad peak (60 to 110 °C); this could be ascribed to the loss of the surface-bound moisture from this amorphous polymer. Also, the exothermic peak for Ag was at 245 °C due to decomposition of the polymer. The endothermic peaks of PM of drug with Ag demonstrated the characteristic features of polymers and drug alone indicating that there were no interactions between the polymers and KT. Thermograms for Ch-Ag loaded MPs showed no endothermic peak for the drug. The DSC thermogram of MPs showed complete disappearance of KT; this could be ascribed to complete amorphization of the drug within the MPs. 

### 3.7. FTIR 

Figure 7 displays the FTIR spectra of KT, Ag, and loaded Ch-Ag based MPs formulations. In the spectrum of KT, characteristic peaks (3340 cm^−1^ ascribed for NH stretch; 1144 cm^−1^ ascribed for carbonyl group (C=O) stretch; and 1556 due to aromatic stretching. There was no discernible detection in the drug’s IR spectra when KT was trapped in Ch-MPs. This peak in MPs was displaced at 3349 cm^−1^, which is an indicator that the KT reaction with Ag was noticeable. KT exhibited a high absorption peak at 3343 cm^−1^ that could be ascribed to NH stretch. Furthermore, KT had two additional strong peaks at 1560 cm^−1^ and 1146 cm^−1^, which were also obtained in loaded MPs at the exact same location. The amide or the hydroxyl groups of CS cannot form strong connections with the amino group of KT, according to these data. In order to prevent any drug-polymer interactions from occurring during the creation of MPs, FTIR investigations were conducted [36]. 

The O-H bond stretching vibrations of alginate were seen in the Ag FT-IR spectra between 3000 and 3600 cm^−1^. As previously mentioned, bands at 1595 cm^−1^ and 1406 cm^−1^ were identified as the symmetric and asymmetric stretching peaks of carboxylate salt (COO-) groups, respectively. On the other hand, it was seen in the FT-IR spectrum for KT-laden MPs that the drug’s distinctive absorption peak (Ch0.5:Ag0.5:KT10) occurred in the loaded MPs, which is likely evidence that KT molecules were filled in the polymeric network. A symmetric stretching for carboxyl groups shifted from 1595 cm^−1^ to 1570 cm^−1^ and the symmetric stretching of the carboxylate group shifted slightly to 1406 cm^−1^, it was also noticed [20].

### 3.8. In-Vitro Release Study of Chitosan-Sodium Alginate MPs

The cumulative release of KT from Ch-Ag MPs in PBS at 35 °C is shown in Figure 8. After optimizing the conditions, only two different ratios from all prepared KT loaded Ch-Ag MPs were selected for the in-vitro release study depending on size, zeta potential, EE% and different formulation parameters observed during the formulation process. Figure 9 shows the sustained release profile of KT from the Ch-Ag MPs up to 8 h. Furthermore, the MPs redispersed in PBS (pH 7.4) after centrifugation containing an amount of drug equivalent to 5 mg/mL, showed a significant (*p* < 0.05) controlled drug release pattern compared with the control KT solution. KT released completely from the drug solution after 3 h, while that from the selected MPs was approximately 92% and 78% after 8 h from (Ch1:Ag0.5:KT10) and (Ch0.5:Ag0.5:KT10), respectively. The presented data indicated that the pattern of release of KT from loaded MPs was nearly the same. However, increasing Ch concentrations led to decreases in drug release rates. This could be due to the fact that high concentrations of Ch cause the media to be more viscous and thus hinder the release of KT; this could explain nearly 14% more decreases in the release pattern of KT compared with a smaller ratio for the same period of time [26]. The results were in agreement with those reported for KT release from polymeric micelles, with 60% of the drug released after a period of 8 h [36,37].

Figure 9 shows the release profile of KT from Ch-Ag MPs prepared using the pre-gel method. The release profile of KT from these MPs did not differ significantly from that of the above data. Both MPs were prepared using the same polymers, except that the method of preparation was different, as this method involved pre-gelation of Ag with CaCl2 before the formulation. At a constant Ag concentration, increasing concentrations of Ch led to a decrease in drug release rate as 100% and 87% of KT released after 8 h for ratios (Ch1:Ag1:KT10) and (Ch1.5:Ag1:KT10), respectively. Moreover, Figure 10 indicates a nearly 2-fold decrease in the release profile of KT from loaded MPs of ratio (Ch1.5:Ag1:KT10) after 2 h compared with that of ratio (Ch1:Ag1:KT10). This indicated the effect of Ch concentrations on the release pattern of the formed MPs. Both Ch and Ag are hydrophilic polymers, so the release medium required more time to hydrate in order to penetrate into the polymeric matrix of MPs and release the entrapped drug. The rate of hydration and release is likely to depend on the concentrations of cross-linkers and the porosity of MPs.

### 3.9. Mucoadhesion Investigation

The stability and mucoadhesion forces of Ch- MPs with mucin were studied by determining the viscosity of mucin solution after incubation with KT loaded Ch-Ag MPs for 2 h at 35 °C. The viscosity was compared with that of Ch solution incubated at the same conditions.

For both formulations prepared using the ionic gelation method and the ionic pre-gelation method, Table 11 shows the measured viscosity of these formulations before and after incubation. The results did not show significant (*p* > 0.05) changes in the amount measured. For example, before incubation the viscosity for Ch1: Ag0.5:KT10 MPs was 3.17 ± 0.057 mPa.s but after 2 h changed to 3.08 ± 0.072 mPa.s at 35 °C, respectively. On the other hand, for Ch1.5:Ag1:KT10 MPs prepared using the pre-gelation method as there was a significant (*p* < 0.05) decrease in the measured viscosity after 2 h incubation with mucin dispersion compared with viscosity value before incubation. Table 11 shows that the viscosity before and after 2 h were 4.80 ± 0.60 mPa.s and 3.40 ± 0.083 mPa.s. This could be attributed to possible interactions of the negatively charged mucin with the cationic Ch or that of calcium ions, which may be present free in solution.

From these findings, it could be suggested that there were weak interactions between the investigated MPs and mucin, but Ch0.5:Ag0.5:KT10 NP significantly altered the viscosity of mucin dispersion, and the remarkable decrease in the viscosity values did not favor the use of these formulations for further work.

Comparing the zeta potential of the mucin colloidal system before and after incubation, the results showed slight decreases in the zeta potential for (mucin and Ch solution) and (mucin and Ch-Ag MPs). Figure 10 shows the zeta potential values of 66 mV and 61.5 mV for the combination of Ch solution and mucin before and after incubation, respectively.The results obtained with Ch1: Ag0.05:KT10 MPs were 28.7 ± 0.54 mV and 25.7 ± 0.34 mV before and after incubation, respectively. However, for Ch1.5:Ag1:KT10 MPs prepared using the pre-gelation method the decrease in the zeta potential was significant (*p* < 0.05) where zeta at zero time was 13.8 ± 0.78 mV and after 2 h became 10.87 ± 0.798 mV, Figure 10. This insignificant (*p* < 0.05) reduction could be related to electrostatic interactions of the negatively charged sialic groups of mucin with the positively charged amino groups of Ch. These findings indicated that the prepared Ch0.5:Ag0.05:KT10 formulation could interact with mucin with no alteration in corneal viscosity [22]. 

### 3.10. Ex-Vivo Permeation Study for Chitosan-Based MP Formulations

The same tested MPs formulations used in the previous experiment were used here in this study for the same reasons mentioned before. Figure 11 shows the corneal permeation of the drug from Ch-Ag based MPs. The permeation parameters were calculated: flux, P_app_ and lag time (t_L_), Table 12. It is clear from Figure 11 that the amount of KT permeated through the cornea from KT solution is higher than that from loaded MPs. The permeation parameters for this formulation (Ch01:Ag0.5:KT10) including steady-state flux, apparent permeability coefficient (P_app_) were 91.42 ± 2.1 (µg/h) and 2.97 ± 0.71 (cm/sec), respectively, which is obviously lower compared with the same data for KT solution, Table 12. 

The Franz–diffusion cell model is a static model, while in the real in-vivo setting of the eye, there will be many different factors such as tear turnover, draining of the instilled solution from the eye, and blinking processes that affect the amount of drug that reaches the target site [23]. However, in the case of KT-loaded MPs, it can be argued that the recorded decreases in flux from MPs compared to the control solution are likely to be compensated by prolonging the precorneal residence time, viscosity, and mucoadhesion. Meanwhile, for the Ch1:Ag0.5:KT10 formulation, the release pattern of KT from this formulation was relatively lower compared with the KT solution (Figure 11). This argument has been demonstrated in in-vivo rabbit models [38].

Furthermore, the calculated t_L_ for Ch: Ag MPs were 0.383 ± 0.18 h and 0.54 ± 0.44 h which is significantly (*p* < 0.05) longer compared with drug solution. These results could be attributed to the fact that the drug takes a long time to release from these MPs compared to the control KT solution, which demonstrated a higher extent and rate of permeation across the excised cornea compared with Ch-Ag MPs. These findings, which were in line with the in-vitro release data, indicated that KT’s ability to permeate the cornea was influenced by its release properties and that the rate of permeation through the cornea was the rate-limiting factor.

### 3.11. MTT Cytotoxcity Assay

#### 3.11.1. MTT Assay on Human Lens Cells

Both KT-free and KT-loaded MPs were employed in this study. Figure 12 shows images of the lens cell one hour before and after treatment with KT solution and Ch1:Ag0.5:KT10 MPs. It is evident that the number of cells available decreased after 1 h post-treatment of the tested substance, which reflects the effect of the prepared formulation on the cell line. It is also worth mentioning that control wells were incubated with growth media without test solution, and BKC (0.01% *w/v*) was used as an example of a positive control.

Figure 13 indicates the percent control measured for cells after 4 h of treatment with MTT. It could be seen that the cytotoxicity of Ch: Ag MPs for both formulations tested was insignificant. Upon comparing the percent cell viability of Ch1:Ag0.5:KT10 with the drug solution, there was a significant difference (*p* < 0.05) between them. This could indicate that Ch polymer is a potential ingredient in MP preparation as it may be responsible for samples’ cytotoxicity. However, there was an insignificant difference between the formulations, both plain and medicated.

#### 3.11.2. MTT Assay on Human Primary Corneal Epithelium Cells

Different types of ocular cells were used here to test the cytotoxicity of selected MPs, both drug-free and loaded with KT.

Figure 14 shows the estimated cell viability for the investigated samples and the positive strong irritant control (BKC). The percentage cell viability recorded for KT solution and MPs was significantly (*p* < 0.05) greater compared to the positive control. On the contrary, there were no significant changes in cell viability among KT solutions and MPs. However, a significant (*p* < 0.05) difference was recorded for KT-loaded MPs loaded with KT solution. According to regulatory standards for ophthalmic pharmaceuticals, these results reveal that KT-loaded MPs exhibit an acceptable percentage of cell survival when compared to control samples, as illustrated in Figure 14. Therefore, taking into account the dynamics of the tears and the length of time those MPs are likely to stay in the eye, the evaluated formulations were deemed appropriate for ocular use.

## 4. Conclusions

Hybrid Ag-Ch MPs were successfully prepared and characterized; KT-loaded MPs demonstrated superior mucoadhesion through interactions with the in-vitro mucin model. The ex-vivo corneal permeation study demonstrated that the hybrid Ag-Ch MPs studied had the capacity to sustain drug permeation. These research findings evidently showed the prominence of Ag-Ch-based MPs in delivering KT to the ocular tissue. MTT cytotoxicity assays on human primary corneal epithelial cells as well as human lens cells revealed that all KT-loaded MP formulations demonstrated significantly higher and acceptable cell viability (%) compared with that recorded for the strong irritant control. In conclusion, this study has presented the hybrid Ag-Ch MPs as promising polymeric carriers for the ocular delivery of KT with improved safety and sustained ocular delivery.

## Figures and Tables

**Figure 1 polymers-15-02773-f001:**
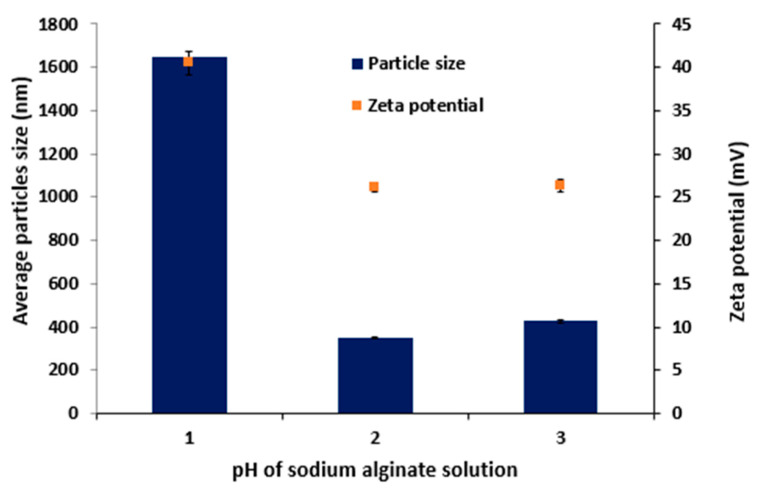
Effects of changing pH of sodium alginate solution on the average particle sizes and zeta potential of Ch1.5:Ag1 MPs prepared using the ionic pre-gelation method, CaCl_2_ (0.22% *w*/*v*), volume ratio (1:1) *v*/*v*, Ch (Ph = 5.5). Results are expressed as means ± SD, n = 3.

**Figure 2 polymers-15-02773-f002:**
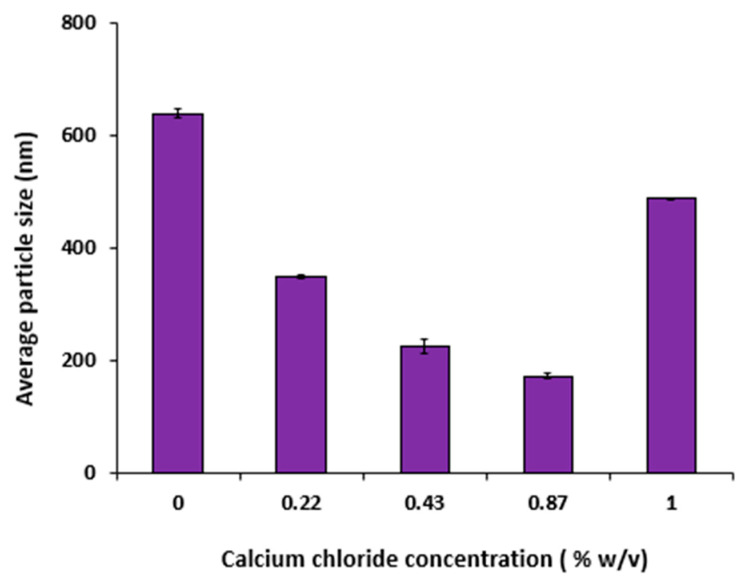
Effects of different concentrations of calcium chloride on the average particles’ sizes of Ch1.5:Ag1 formulation prepared using the ionic pregelation method. Results expressed as mean values ± SD, (n = 3).

**Figure 3 polymers-15-02773-f003:**
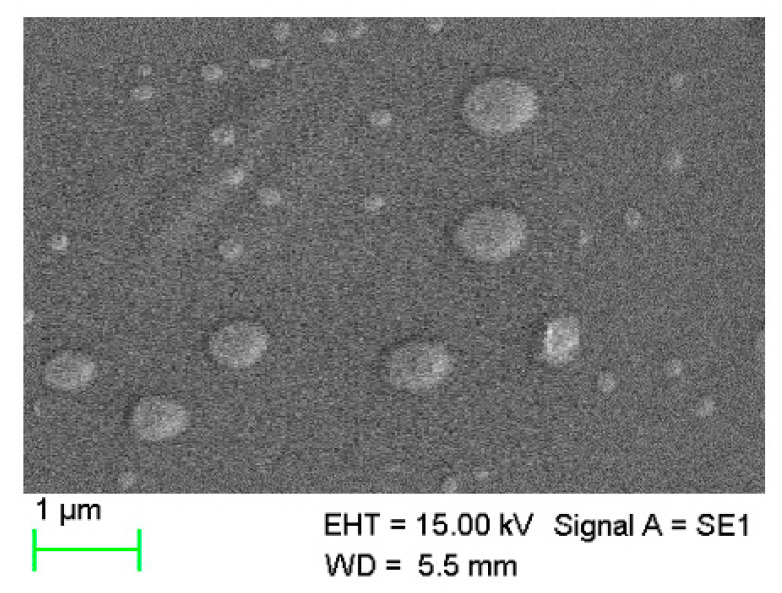
Scanning electron microscopic (SEM) image for (Ch1:Ag0.5:KT10) loaded MPs.

**Figure 4 polymers-15-02773-f004:**
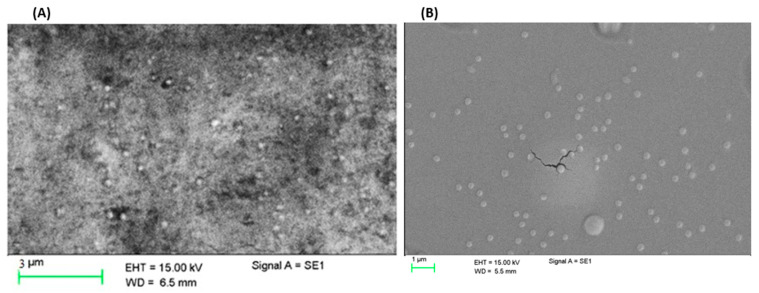
Scanning electron microscopic (SEM) image for plain (Ch1.5:Ag1:KT10) (**A**), and loaded (**B**) MPs prepared using the pre-gel method.

**Figure 5 polymers-15-02773-f005:**
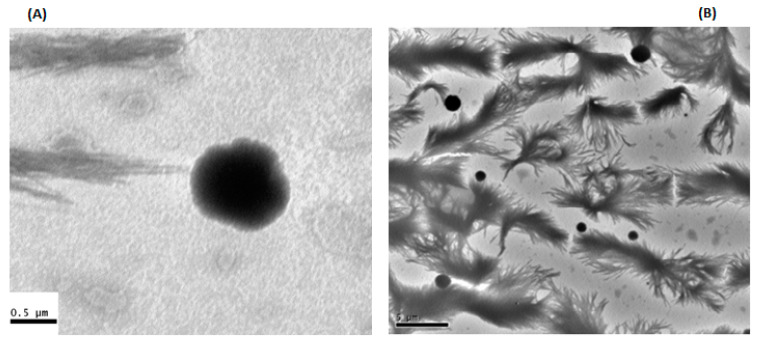
Transmission electron microscopic (TEM) image for (**A**) Ch1:Ag0.5:KT10 and (**B**) (Ch1.5:Ag1: KT10) loaded MPs prepared using the ionic gelation and ionic pregelation methods, respectively.

**Figure 6 polymers-15-02773-f006:**
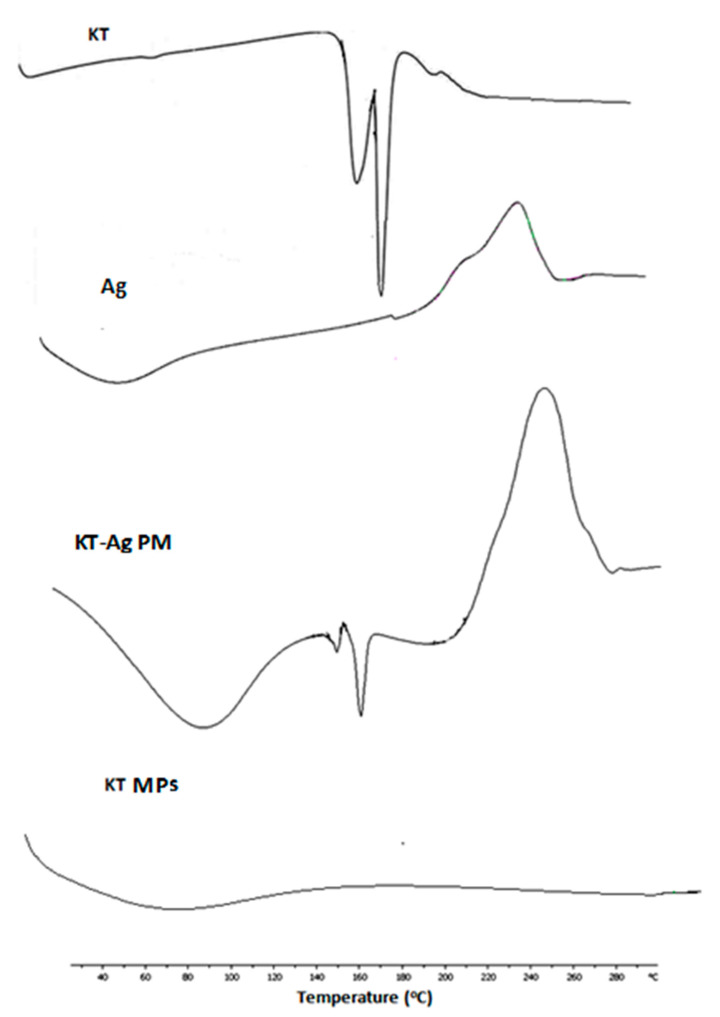
DSC thermograms for ketorolac tromethamine (KT), sodium alginate (Ag), their physical mixture (KT-Ag PM) and Ch-Ag loaded MPs.

**Figure 7 polymers-15-02773-f007:**
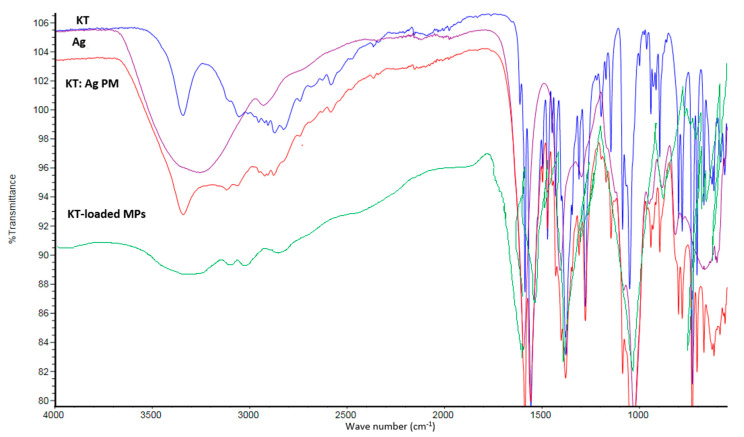
FT-IR spectra for ketorolac tromethamine (KT), Sodium alginate (Ag), their physical mixture (KT-Ag PM) and loaded MPs.

**Figure 8 polymers-15-02773-f008:**
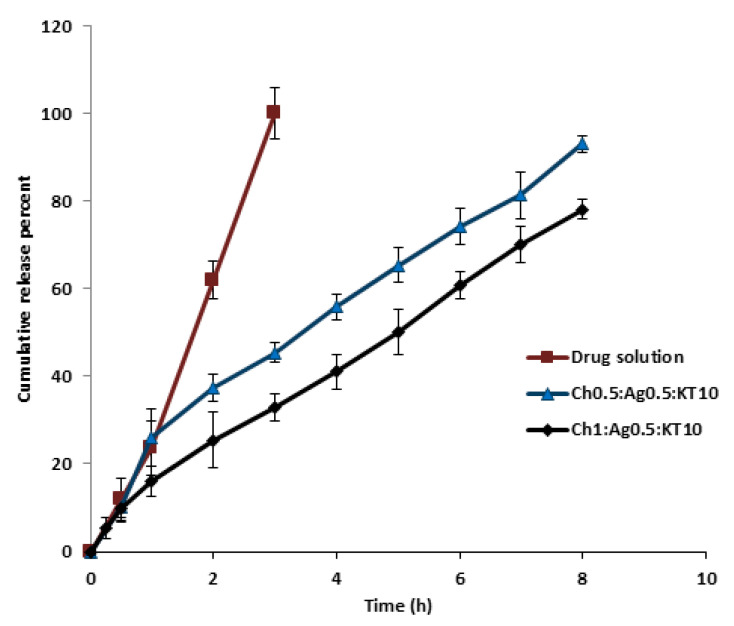
In-vitro release profile of ketorolac tromethamine (KT) from KT solution and loaded MPs with different Ch: Ag (mg/mL) mass ratio prepared using the ionic-gelation method. Results are expressed as means ± SD, n = 3.

**Figure 9 polymers-15-02773-f009:**
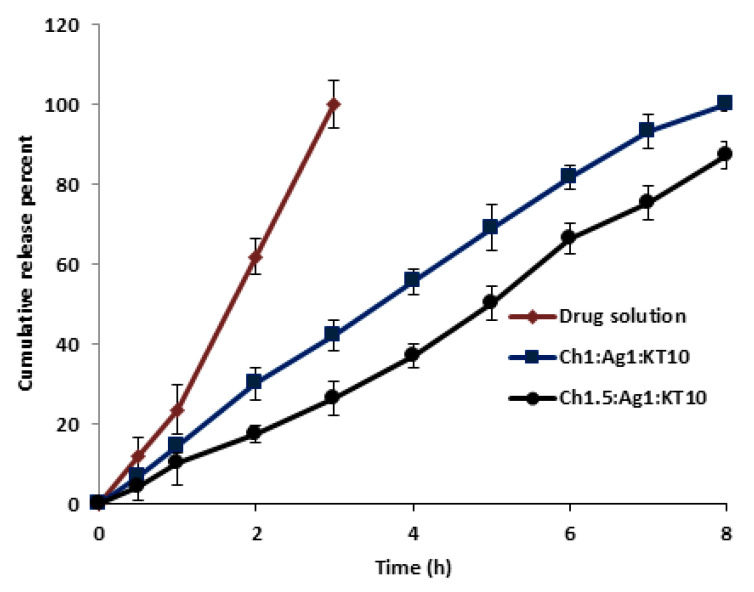
In-vitro release profiles of ketorolac tromethamine (KT) from KT solution and loaded MPs with different Ch:Ag:KT10 (mg/mL) mass ratio prepared using the pre-gelation method. Results are expressed as means ± SD, n = 3.

**Figure 10 polymers-15-02773-f010:**
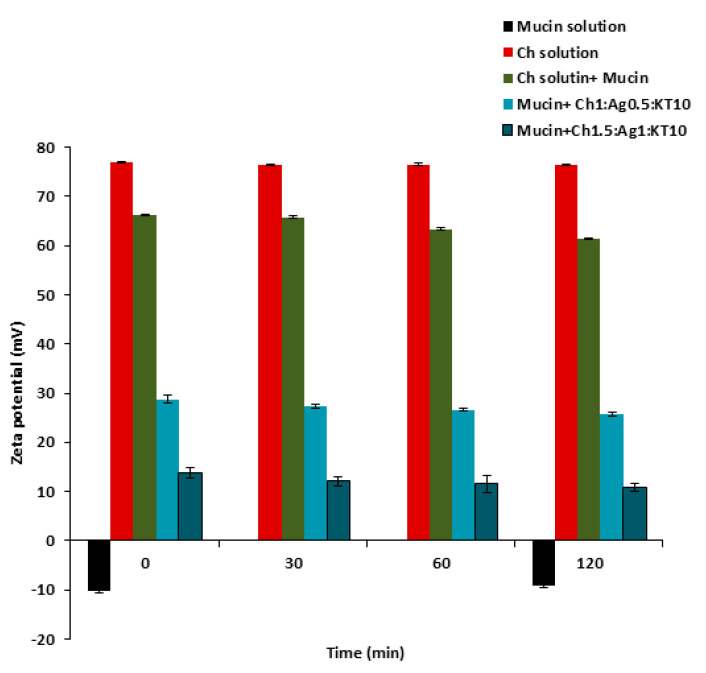
Zeta potential of (■) mucin solution (0.4mg/mL), (■) Ch solution alone, (■) Ch solution + mucin, (■) mucin + Ch1:Ag0.5:KT10 MPs, and (■) mucin + Ch1.5:Ag1:KT10 MPs before and after incubation at different time intervals. Results are expressed as means ± SD, n=3.

**Figure 11 polymers-15-02773-f011:**
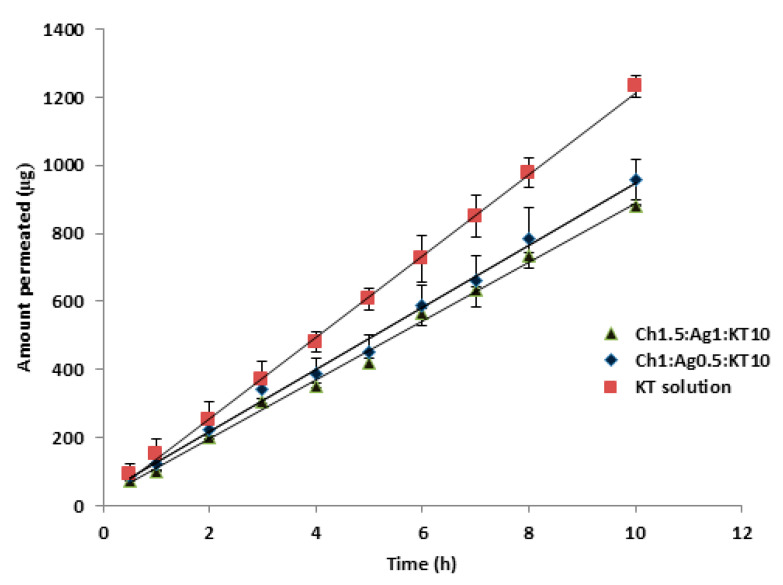
Transcorneal permeation profiles of ketorolac tromethamine (KT) from KT solution and drug loaded-MPs (Ch-Ag) using excised porcine corneas. Results are expressed as means ± SD, n = 3.

**Figure 12 polymers-15-02773-f012:**
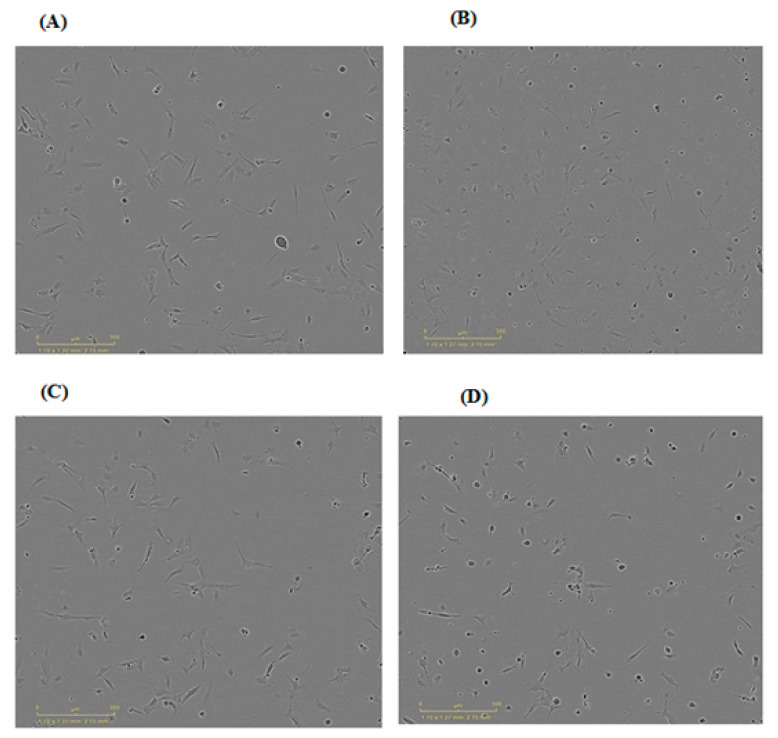
Morphology of human lens cells 1 h before treatment (left side) and 1 h after treatment (right side). (**A**) 1 h before treatment with KT-loaded PPs formulation (Ch1:Ag0.5: KT10), (**C**) 1 h after treatment with the same formulation, (**B**) 1 h before treatment with KT solution (5 mg/mL) and (**D**) 1 h after treatment with the KT solution.

**Figure 13 polymers-15-02773-f013:**
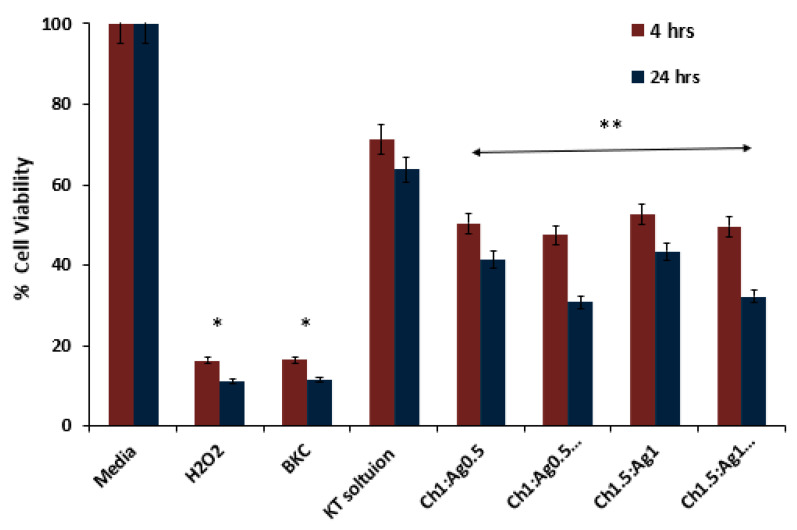
MTT assay results on human lens cells showing percent cell viability 4 h after treatment with KT solution (5 mg/mL), drug free and loaded MP formulations. Results are expressed as means ± SD, n = 3. * indicated significant differences (*p* < 0.05); ** indicated non-significant differences (*p* > 0.05).

**Figure 14 polymers-15-02773-f014:**
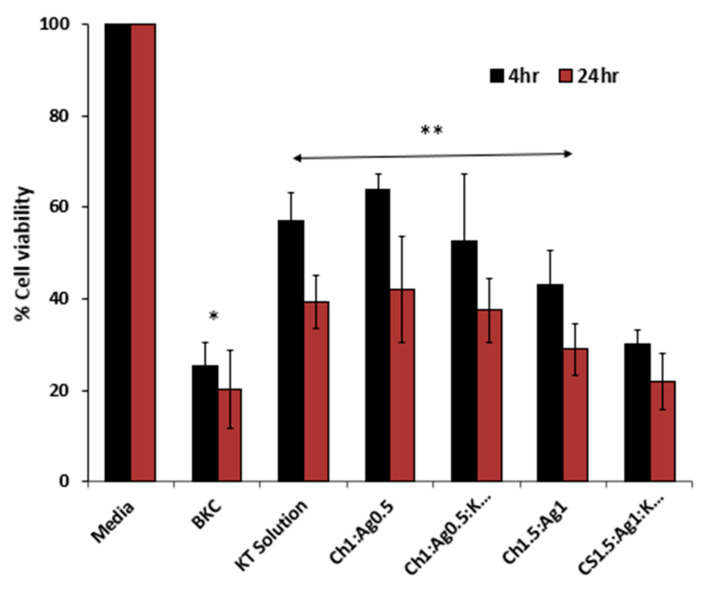
MTT assay results on human primary corneal epithelial cells after 4 h and 24 h treatment with plain and loaded alginate MP formulations. Results are expressed as means ± SD, n = 3. * indicates significant differences (*p* < 0.05); ** indicated non-significant differences (*p* > 0.05).

**Table 1 polymers-15-02773-t001:** Effects of changing chitosan concentrations on the average particle sizes of chitosan-sodium alginate (Ch-Ag) MPs prepared using ionic gelation method. Results are expressed as mean values ± SD, (n = 3).

Formulation Code	Ch (%) *w*/*v*	Ag (%) *w*/*v*	Particle Size (nm)	Zeta Potential (mV)	PDI	Visual Observation
Ch0.3:Ag0.5	0.03	0.05	1322.4 ± 18.4	19.3 ± 0.21	0.884 ± 0.080	Aggregates
Ch0.4:Ag0.5	0.04	0.05	472.1 ± 3.70	25.96 ± 2.44	0.413 ± 0.041	Opalescent solution
Ch0.5:Ag0.5	0.05	0.05	402.3 ± 4.52	32.53 ± 1.34	0.230 ± 0.092	Opalescent solution
Ch0.6:Ag0.5	0.06	0.05	405.3 ± 8.63	32.6 ± 0.808	0.249 ± 0.053	Opalescent solution
Ch0.8:Ag0.5	0.08	0.05	396.2 ± 6.40	36.83 ± 1.24	0.274 ± 0.021	Opalescent solution
Ch1:Ag0.5	0.1	0.05	462.1 ± 9.90	34.1 ± 0.32	0.306 ± 0.102	Opalescent solution
Ch2:Ag0.5	0.2	0.05	524.2 ± 11.10	38.7 ± 1.01	0.446 ± 0.102	Opalescent solution

**Table 2 polymers-15-02773-t002:** Effects of changing chitosan concentrations on the average particle sizes of chitosan-sodium alginate (Ch-Ag) MPs prepared using ionic gelation method. Results are expressed as means ± SD, n = 3.

Formulation Code	Ch (%) *w*/*v*	Ag (%) *w*/*v*	particle Size (nm)	Zeta Potential (mV)	PDI	Visual Observation
Ch0.3:Ag0.75	0.03	0.075	1322 ± 10.5	−1.56 ± 0.50	1.00 ± 0.098	Aggregates
Ch0.4:Ag0.75	0.04	0.075	1438 ± 8.6	25.2 ± 3.46	0.808 ± 0.014	Aggregates
Ch0.5:Ag0.75	0.05	0.075	508.4 ± 27.8	32.6 ± 1.11	0.266 ± 0.003	Opalescent solution
Ch0.6:Ag0.75	0.06	0.075	522.13 ± 1.5	32.36 ± 0.60	0.344 ± 0.021	Opalescent solution
Ch0.8:Ag0.75	0.08	0.075	596.9 ± 10.6	32.06 ± 0.67	0.357 ± 0.042	Opalescent solution
Ch1:Ag0.75	0.1	0.075	643.23 ± 3.7	35.30 ± 0.25	0.117 ± 0.082	Opalescent solution
Ch2:Ag0.75	0.2	0.075	1677 ± 23.1	18.98 ± 0.32	1.00 ± 0.076	Aggregates

**Table 3 polymers-15-02773-t003:** Particle sizes and zeta potential measured for Ch-Ag MPs prepared at different (Ch-Ag) volume ratios (*v*/*v*) and prepared using the ionic gelation method. Results are expressed as means ± SD, n = 3.

Ch:Ag Ratio (*v*/*v*)	Ch0.5:Ag0.5		Ch1:Ag0.5		Ch2:Ag0.5	
	Particle Size (nm)	Zeta Potential (mV)	Particle Size (nm)	Zeta Potential (mV)	Particle Size (nm)	Zeta Potential (mV)
1:1	1344 ± 31.95 (PDI: 0.974 ± 0.189)	−19.3 ± 0.264	534.8 ± 11.3 (PDI: 0.561 ± 0.040)	27.8 ±1.22	565.2 ± 15.16 (PDI: 0.339 ± 0.002)	34.1 ± 0.10
2:1	394.6 ± 2.628 (PDI: 0.254 ± 0.098)	32.5 ± 0.208	434.9 ± 14.94 (PDI: 0.306 ± 0.037)	33.7 ± 0.88	528.36 ± 22.72 (PDI: 0.373 ± 0.280)	38.0 ± 0.50
2.5:1	402.3 ± 4.52 (PDI: 0.230 ± 0.092)	32.5 ± 1.34	462.1 ± 9.9 (PDI: 0.306 ± 0.102)	34.1 ± 0.32	524.2 ± 11.10 (PDI: 0.446 ± 0.102)	38.7 ± 1.01
3:1	384.6 ± 18.81 (PDI: 0.052 ± 0.011)	29.4 ± 0.51	478.1 ± 7.01 (PDI: 0.245 ± 0.099)	30.5 ± 1.44	489.3 ± 19.91 (PDI: 0.650 ± 0.190	37.9 ± 0.55

**Table 4 polymers-15-02773-t004:** Particle sizes recorded for Ch-Ag MPs prepared at different amount of Tween 80 prepared using the ionic gelation method. Results are expressed as means ± SD, n = 3 *.

Tween 80 (%) *w*/*v*	Ch0.5:Ag0.5	Ch1:Ag0.5	Ch2:Ag0.5
0	394.6 ± 2.628(PDI: 0.254)	434.9 ± 14.94 (PDI: 0.306)	528.36 ± 22.72 (PDI:0.373)
0.5	343.93 ± 5.13 (PDI: 0.115)	305.23 ± 10.65 (PDI: 0.228)	461.83 ± 6.76 (PDI: 0.334)
1.0	340.11 ± 6.88 (PDI: 0.344)	320.77 ± 2.91 (PDI: 0.222)	438.51 ± 14.4 (PDI: 0.129)

* Ch:Ag volume ratio (2.5:1 *v*/*v*).

**Table 5 polymers-15-02773-t005:** Particle sizes, zeta potential and EE% of KT loaded Ch-Ag MPs. Results are expressed as means ± SD, n = 3 *.

Formulation Code	Average Particle Size (nm)	Zeta Potential (mV)	PDI	EE%
Ch0.5:Ag0.5	497.90 ± 10.55	26.7 ± 2.34	0.175 ± 0.03	24.53 ± 2.48
Ch1:Ag0.5	525.33 ± 11.87	32.1 ± 3.76	0.233 ± 0.01	21.70± 1.09
Ch2:Ag0.5	812.7 ± 13.76	36.6 ± 1.56	0.656 ± 0.126	19.26 ± 1.33

* Ch:Ag volume ratio (2.5:1 *v*/*v*), Tween 80 (0.5%) *w*/*v*.

**Table 6 polymers-15-02773-t006:** Effects of different concentrations of KT on the EE% of Ch-Ag MPs. Results are expressed as means ± SD, n = 3.

KT Conc. (mg/mL)	Ch0.5:Ag0.5				Ch1:Ag0.5			
	Size (nm)	Zeta Potential (mV)	PDI	EE%	Size (nm)	Zeta Potential (mV)	PDI	EE%
5	497.9 ± 10.55	26.7 ± 2.34	0.175 ± 0.03	24.53 ± 2.48	525.3 ± 11.87	32.1 ± 3.76	0.233 ± 0.01	21.70 ± 1.09
10	523.3 ± 5.76	24.7 ± 0.88	0.434 ± 0.02	69.5 ± 2.13	679.7 ± 3.76	28.7 ± 0.54	0.145 ± 0.02	55.7 ± 12.67
15	1287 ± 0.843	13.7 ± 4.98	0.854 ± 0.09	Drug * PPT	1388 ± 0.722	18.9 ± 2.09	1.00 ± 0.21	Drug PPT *

* Ch: Ag volume ratio (2.5:1 *v*/*v*), Tween 80 (0.5%) *w*/*v*. * formulation containing KT (15 mg/mL), drug precipitates (PPT).

**Table 7 polymers-15-02773-t007:** Particle sizes, PDI and zeta potential of Ch-Ag MPs prepared using the ionic pregelation method. * Results are expressed as means ± SD, n = 3.

Formulation Code	Ch (*w*/*w*)	Ag (*w*/*w*)	Size (nm)	Zeta Potential (mV)	PDI
Ch1:Ag1	1	1	449.3 ± 12.5	41.43 ± 3.64	0.109 ± 0.036
Ch1.5:Ag1	1.5	1	348.8 ± 2.0	26.13 ± 0.47	0.121 ± 0.005
Ch3:Ag1	3	1	1644.0 ± 44.5	33.93 ± 2.01	0.293 ± 0.229
Ch5:Ag1	5	1	1867.2 ± 36.2	45.56 ± 1.76	0.718 ± 0.067
Ch1:Ag1.5	1	1.5	1677.2 ± 21.4	32.43 ± 2.76	0.810 ± 0.017
Ch1:Ag3	1	3	1876.6 ± 17.2	33.56 ± 2.22	0.847 ± 0.264
Ch1:Ag5	1	5	2010.3 ± 26.6	35.06 ± 0.55	0.862 ± 0.025

* Sodium alginate solution (pH = 5.3) and Ch solution (pH = 5.5), CaCl_2_ concentration (0.22%) *w*/*v*, volume ratio of Ch:Ag (1:1 *v*/*v*).

**Table 8 polymers-15-02773-t008:** Effects of different concentrations of calcium chloride on the sizes of Ch-Ag MPs prepared using the ionic pregelation method. * Results are expressed as means ± SD, n = 3.

CaCl_2_ Concentration (% *w*/*v*)	Ch1.5:Ag1		
	Particle Size (nm)	Zeta Potential (mV)	PDI
0	639.8 ± 7.89	29.12 ± 0.33	0.428± 0.171
0.1	487.4 ± 1.90	25.3 ± 1.85	0.334 ± 0.025
0.22	348.8 ± 2.00	26.13 ± 0.47	0.121 ± 0.005
0.43	225.5 ± 11.22	16.4 ± 3.60	0.273 ± 0.382
0.87	172.9 ± 4.08	5.7 ± 1.13	0.288 ± 0.292

* Ag solution (pH = 5.3) and Ch solution (pH = 5.5), volume ratio of Ch:Ag (1:1 *v*/*v*).

**Table 9 polymers-15-02773-t009:** Effects of stirring time on the sizes of Ch-Ag MPs prepared using the ionic pregelation method. Results are expressed as mean ± SD, n = 3 *.

Stirring Time (h)	Ch1.5:Ag1		
	Particle Size (nm)	Zeta Potential (mV)	PDI
0.5	348.8 ± 2.0	26.13 ± 0.47	0.121 ± 0.005
1	361.5 ± 5.6	24.1 ± 0.46	0.31 ± 0.061
2	403.5 ± 34.3	20.9 ± 1.18	0.232 ± 0.013
5	461.2 ± 18.5	20.3 ± 0.46	0.499 ± 0.116
8	480.6 ± 20.7	19.2 ± 1.26	0.502 ± 0.052

* Ag solution (pH = 5.3) and Ch solution (pH = 5.5), CaCl_2_ concentration (0.22% *w*/*v*) and volume ratio of Ch:Ag (1:1 *v*/*v*).

**Table 10 polymers-15-02773-t010:** Effects of different concentrations of KT (mg/mL) on the size, zeta potential and EE% of chitosan –sodium alginate (Ch-Ag) MPs prepared using the pre-gel method. Results are expressed as mean values ± SD, (n = 3) *.

KT Conc. (mg/mL)	Ch1.5:Ag1				Ch1:Ag1			
	Particle Size (nm)	Zeta Potential (mV)	PDI	EE%	Particle Size (nm)	Zeta Potential (mV)	PDI	EE%
5	371.6 ± 3.06	20.5 ± 1.08	0.111 ± 0.04	43.12 ± 3.55	477.2 ± 2.32	30.5 ± 2.06	0.255 ± 0.10	51.44 ± 5.33
10	620.1 ± 7.22	13.8 ± 0.78	0.422 ± 0.05	81.22 ± 2.17	513.6 ± 4.21	10.7 ± 1.03	0.311± 0.09	89.09 ± 1.66

* Ch solution (pH = 5.5), Ag solution (pH = 5.3), CaCl_2_ concentration (0.22% *w*/*v*), stirring time (0.5 h).

**Table 11 polymers-15-02773-t011:** Viscosity values (mPa.s) of mucin dispersion before and after incubation with Ch solution and Ag-Ch MPs prepared using both ionic gelation and pregelation methods. Results are expressed as means ± SD, n = 3.

Time (min)	Mucin + Ch Solution	Mucin + Ch0.5:Ag0.5:KT10	Mucin + Ch1:Ag0.5:KT10	Mucin + Ch1.5:Ag1:KT10
0	5.93 ± 0.115	2.77 ± 0.120	3.17 ± 0.057	4.80 ± 0.600
5	5.92 ± 0.064 *	2.60 ± 0.105	3.16 ± 0.111	4.66 ± 0.165
15	5.77 ± 0.058	2.43 ± 0.115	3.13 ± 0.004	4.58 ± 0.210
30	5.35 ± 0.136	2.29 ± 0.241	3.11 ± 0.011	4.31 ± 0.271
60	5.00 ± 0.100	2.88 ± 0.322	3.09 ± 0.064	3.90 ± 0.519
120	4.97 ± 0.116 *	1.75 ± 0.155	3.08 ± 0.072	3.40 ± 0.083

* Significant difference (*p* < 0.001) between viscosity values for (mucin + Ag-Ch MPs) and (mucin + Ch solution) at 5 min and at 120 min.

**Table 12 polymers-15-02773-t012:** Flux, apparent Permeability Coefficient (P_app_), and lag time (t_L_) of KT. Results are expressed as means ± SD, n = 3.

Formulation Code	Flux (µg/h)	P_app_ × 10^−6^ (cm/s)	t_L_ (h)
KT-solution	119.60 ± 0.96 *	5.31 ± 0.13	0.198 ± 0.34
Ch1:Ag0.5:KT10	91.42 ± 2.1	2.97 ± 0.71	0.383 ± 0.18
CS1.5:SA1:KT10	79.11 ± 1.22	2.33 ± 0.93	0.540 ± 0.44

* Signiant difference (*p* < 0.001) between the measured parameters for KT solution compared with that of all other tested drug loaded formulations.

## Data Availability

Not applicable.

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
