# Peer review of "Formulation and In-Vitro/Ex-Vivo Characterization of Pregelled Hybrid Alginate–Chitosan Microparticles for Ocular Delivery of Ketorolac Tromethamine"

_polymers, 2023, doi:10.3390/polym15132773_

Round 1
Reviewer 1 Report
The present research manuscript entitled “Formulation and in vitro / ex-vivo characterization of pregelled hybrid alginate–chitosan nanoparticles for ocular delivery of ketorolac tromethamine” by Fathalla et al. is novel and well-researched. The researchers conducted extensive in vitro experiments to prove their claim. The results are interesting, the data representation is excellent, and the flow of the manuscript is very good. Overall, the quality of the manuscript is very good. However, this manuscript needs revision and my comments are as follows.
Comment 1. The researchers used UV-Spectrophotometer to determine the entrapment efficiency and drug release. In my opinion, these parameters should be determined by using HPLC/LCMS.
Comment 2. What about the toxicity of the nanoparticles? The researchers can conduct any toxic potential of the nanoparticles on goat eye that can easily available in the market.
Comment 3. What about the irritation potential of the nanoparticles? A Hen egg test-chorioallantoic membrane (HET-CAM) test should be conducted to improve the overall quality of the present investigation.
Comment 4. The researchers should implicate statistical analysis in Figures 13 and 14.
Comment 5. The authors used many non-scientific languages/words in the manuscript like generation of nanoparticles and many more. Overall, the language quality is poor. Kindly improve the language of the manuscript from top to bottom.
I find many grammatical errors throughout in the manuscript. Kindly revise the manuscript from top to bottom.
Author Response
Dear Reviewer,
I would like to take the opportunity to thank you for your valuable comments. We have carefully considered your comments and point-to-point responses are provided as below:
The present research manuscript entitled “Formulation and in vitro / ex-vivo characterization of pregelled hybrid alginate–chitosan nanoparticles for ocular delivery of ketorolac tromethamine” by Fathalla et al. is novel and well-researched. The researchers conducted extensive in vitro experiments to prove their claim. The results are interesting, the data representation is excellent, and the flow of the manuscript is very good. Overall, the quality of the manuscript is very good. However, this manuscript needs revision and my comments are as follows.
Response: the authors are very delighted to hear that the work has been of interest to the reviewer.
Comment 1. The researchers used UV-Spectrophotometer to determine the entrapment efficiency and drug release. In my opinion, these parameters should be determined by using HPLC/LCMS.
The spectrophotometric method was sensitive, validated and it was devoid of interference from other excipients. Further, spectrophotometry is a simple method and very suitable for routine analysis of large samples. Nevertheless, a stability indicating assay was developed and validated but this was beyond the scope of this manuscript and is sent out for publication.
Comment 2. What about the toxicity of the nanoparticles? The researchers can conduct any toxic potential of the nanoparticles on goat eye that can easily available in the market.
BCOP assay was employed for preliminary screening but data were not shown and the results correlated well with that obtained from cytotoxicity data.
Comment 3. What about the irritation potential of the nanoparticles? A Hen egg test-chorioallantoic membrane (HET-CAM) test should be conducted to improve the overall quality of the present investigation.
As the authors mentioned earlier, the manuscript contained many characterization techniques, HET-CAM is an in vitro irritation model to predict the conjunctival irritation; however, cytotoxicity is an in vitro technique and has been correlated well with HET-CAM. The cytotoxicity was performed on two cell lines to document the irritation potential of the prepared nanoparticles.
Comment 4. The researchers should implicate statistical analysis in Figures 13 and 14.
Figure 13 and Figure 14 have been modified accordingly.
Comment 5. The authors used many non-scientific languages/words in the manuscript like generation of nanoparticles and many more. Overall, the language quality is poor. Kindly improve the language of the manuscript from top to bottom.
The whole manuscript has been now revised and any non-scientific words were corrected.
I find many grammatical errors throughout in the manuscript. Kindly revise the manuscript from top to bottom.
The manuscript has been revised for English and grammar corrections.
Reviewer 2 Report
These research results are interesting for the scientific auditorium, however, the manuscript is not easy to read, and better description and reproducibility in the synthesis and characterization is strongly recommended.
The use of abbreviations in the text must be explained at the first occurrence, such as PLL, NSAIDs, etc.
I am not qualified to assess plagiarism in this document.
My review focuses primarily on materials science to help authors better express their findings.
From the synthesis of the materials to the different evaluations, various parameters such as concentration, pH, ratio, etc. were investigated. So, first, an explanation of the clear nomenclature is required considering all the variations, it will help to follow the authors' strategy in the discussion. The order is also crucial.
What was the basis for selecting only a few nanoparticles for volume variation tests? Why were these pH values selected to evaluate the morphology characteristics?
including schematics showing the interactions that the authors argue will help to understand and correlate with FTIR and other techniques.
The title needs to be redrafted because materials are in the micrometer range, also the SEM images show quasispherical microparticles agglomerated into a larger one. I don't know if the selected images are representative, the size and shape statistics will help.
Why do the authors use TEM to further study the particles, the magnifications and mode of operation do not seem appropriate.
How are alginate and chitosan nanoparticles degraded or evacuated in the ocular system?
The figures need to be enhanced, labeled appropriately, and the information to be visible
Finally, a careful review of the manuscript is recommended, the figure number does not match and some phrases are confusing.
Finally, a careful review of the manuscript is recommended, the figure number does not match and some phrases are confusing.
Author Response
Dear Reviewer,
I would like to take the opportunity to thank you for your valuable comments. We have carefully considered your comments and point-to-point responses are provided as below:
These research results are interesting for the scientific auditorium, however, the manuscript is not easy to read, and better description and reproducibility in the synthesis and characterization is strongly recommended.
The whole manuscript has been revised and edited to remove any ambiguity.
The use of abbreviations in the text must be explained at the first occurrence, such as PLL, NSAIDs, etc.
The full names were given before the first mention of abbreviation.
I am not qualified to assess plagiarism in this document.
The manuscript has been revised and rewritten.
My review focuses primarily on materials science to help authors better express their findings.
From the synthesis of the materials to the different evaluations, various parameters such as concentration, pH, ratio, etc. were investigated. So, first, an explanation of the clear nomenclature is required considering all the variations, it will help to follow the authors' strategy in the discussion. The order is also crucial.
The discussion section has been fully revised and structured with subtitles to allow easy follow up.
What was the basis for selecting only a few nanoparticles for volume variation tests? Why were these pH values selected to evaluate the morphology characteristics?
Nanosize range, zeta potential and highest EE% were the criteria of selection.
including schematics showing the interactions that the authors argue will help to understand and correlate with FTIR and other techniques.
The title needs to be redrafted because materials are in the micrometer range, also the SEM images show quasispherical microparticles agglomerated into a larger one. I don't know if the selected images are representative, the size and shape statistics will help.
The work showed full optimization of processing parameters; therefore some particles were in micron size ranges but those formulation were excluded from further investigation and only NPs with nanosize ranges were studied and were the main focus of the manuscript.
Why do the authors use TEM to further study the particles, the magnifications and mode of operation do not seem appropriate.
Two imaging techniques were employed in an attempt to gain more insights into particles’ morphology and charactersitics.
How are alginate and chitosan nanoparticles degraded or evacuated in the ocular system?
The exhausted NPs or degraded NP should be cleared through nasolacrimal drainage systems by aid of tears turnover from the surface of the eye.
The figures need to be enhanced, labeled appropriately, and the information to be visible.
Figures 1,2, 13 and Figure 14 have been enhanced and replaced accordingly.
Finally, a careful review of the manuscript is recommended, the figure number does not match and some phrases are confusing.
The whole manuscript has been revised and figures numbers were corrected.
Round 2
Reviewer 1 Report
The revision is satisfactory. I don't have further comments.
Author Response
The authors are grateful and delighted to satisfactorily meet the reviewer's comments.
Reviewer 2 Report
Some issues were addressed; however, the manuscript needs further improvements.
As authors show in the manuscript, they do not work with nanosized materials, so it is strongly recommended to correct the description to micromaterials.
The figures need more quality, especially Fig 3, Fig 4, Fig 6, Fig 7 y Fig 12. The figures look like a screen print.
One concern is the reproducibility on the synthesis and other methodologies, authors need to guarantee to be reproduced anywhere.
The authors need to review the paper, they say that figure 8 shows the FTIR spectra, but this does not match., etc.
Majors revisons
Author Response
As authors show in the manuscript, they do not work with nanosized materials, so it is strongly recommended to correct the description to micromaterials.
The title has been changed to microparticles and the word or its abbreviation has been corrected to MPs.
The figures need more quality, especially Fig 3, Fig 4, Fig 6, Fig 7 y Fig 12. The figures look like a screen print.
These figures are instrument/software generated images. These images have passed the quality check by the journal and they are readable.
One concern is the reproducibility on the synthesis and other methodologies, authors need to guarantee to be reproduced anywhere.
The ion gelation methods and pregellation methods have been extensively published before. They are reproducible and can be reproduced by other researchers. However, they require optimization procedures to suit the quality of the materials used and equipment.
The authors need to review the paper, they say that figure 8 shows the FTIR spectra, but this does not match., etc.
It was just a typo and it has been corrected.